# Multiscale Mathematical Analysis of Influencing Factors and Experimental Verification of Microcrack Self-Healing Efficiency of Bitumen Composites Using Microcapsules

**DOI:** 10.3390/ma16145073

**Published:** 2023-07-18

**Authors:** Xin-Yu Wang, Qian Sun, Sai Wang, Rong-Yue Shao, Jun-Feng Su

**Affiliations:** 1School of Mechanical Engineering, Tianjin University of Commerce, Tianjin 300134, China; 2School of Information Engineering, Tianjin University of Commerce, Tianjin 300134, China; 3School of Material Science and Engineering, Tiangong University, Tianjin 300387, China

**Keywords:** self-healing, composite, bitumen, microcapsule, mathematical model, multiscale

## Abstract

The preparation and application of microcapsules containing healing agents have become a crucial way to enhance the self-healing capability of bitumen. This intelligent material has become a hot topic in the field of pavement material and has greatly stimulated the development and applications of pavement engineering. However, there has been no research focused on the relationship of the multistructures from the viewpoint of molecular-size, microsize, and macrosize, which significantly limits the predictions of the self-healing efficiency and structure design of this self-healing material. The purpose of this study was to make a mathematical analysis of the influencing factors of self-healing efficiency based on the self-healing mechanism of bitumen using microcapsules, fully considering the structural dimensions, preparation conditions, and self-healing conditions. In the mathematical analysis, the cross-linking degree of the shell material molecules of the microcapsules was considered for its damage strength from the perspective of molecular structure. The final tip stress of the microcrack was believed to be equal to the puncture strength of the microcapsules in terms of microsize. From a macroscale point of view, the amount of healing agent released from the microcapsule rupture was considered more significant than or equal to the volume of the microcracks. At the same time, the time–temperature superposition principle was applied to simplify the influence factors. The above derivation based on multiscale structures found that the additive amount of the microcapsules, temperature, and time were the three main influencing factors on the self-healing features of bitumen. Finally, the experimental data was investigated considering the three factors, which thoroughly verified the feasibility of the derivation. All results will help to establish a bridge between the initial structural design of self-healing bitumen and the prediction of the final self-healing effects.

## 1. Introduction

Bitumen is one of the pavement materials applied worldwide [1]. Due to exposure to heat, oxygen, and ultraviolet light, its stiffness increases and its relaxation capacity decreases after some years of usage. As an aggregate binder, bitumen also becomes brittle, triggering microcracks [1]. Therefore, intelligent technology is urgently required for application to bitumen, to prolong the service life of the bituminous pavement. Self-healing technology has attracted more and more attention from the field of intelligent material science [2,3,4]. In recent years, self-healing bitumen has attracted interest for the application of this technology in the construction and maintenance of pavements to increase the service time and decrease the maintenance costs [4,5,6]. At the same time, self-healing bitumen can reduce the usage of resources in pavement maintenance, decrease traffic blockage during maintenance, decrease pollutant and greenhouse gas emissions, and elevate the roads’ safety and their lifespan [2].

Nowadays, self-healing bitumen can be divided into intrinsic and extrinsic approaches according to the healing mechanism [6,7]. The inherent approach can be obtained through chemical reactions, including chain-end recombination, molecular bond formation, and photo inducement [8]. The extrinsic approach needs outside aid, including heating or lighting [9]. There are limitations to this approach as it requires the consumption of energy and special equipment. Another extrinsic aid is the help of healing agents incorporated in microcapsules [3,4,5] or microvasculars [8]. Healing agents can be released from broken microcapsules or microvasculars. With the help of healing agents, the crack surfaces are bonded together to achieve self-healing. This process is performed quickly and efficiently, even achieving multi-self-healing capability [8]. Su and Schlangen [10] first reported the self-healing method of bitumen using microcapsules containing oily rejuvenators in 2012. It is still the most mature technology for extrinsic self-healing bitumen in material preparation and application. It is worth mentioning that the world’s first self-healing asphalt pavement using microcapsules containing rejuvenators was built in Tianjin city in China, in 2014 [11]. It was found that the microcracks in this pavement decreased by 75% in five years compared to the nearby pavement without self-healing microcapsules [11]. This material has been widely used in China, becoming a typical commercial self-healing microcapsule material for smart pavements [6,11,12].

Because of the considerable research and commercial value, more and more researchers have invested in the study of this material. The literature research found that the most applied chemical fabrication methods of microcapsules are spraying and coacervation, in situ polymerization, emulsion polymerization, interfacial polymerization, and layer-by-layer deposition [10,11]. Various studies have indicated that the melamine-formaldehyde polymer is the most applied polymeric shell for microcapsule fabrication through in situ polymerization because it can nearly meet all the requirements during the self-healing process [4,10]. It is required that microcapsules must be prepared with good thermal stability to resist the high temperature of bitumen [10]. Moreover, these core/shell structural microcapsules meet specific requirements, including size distribution, and non-biodegradable and mechanical properties [8,10,11,12]. During the chemical polymerization method, the microcapsules’ cost, complexity, and capacity need to be considered for the construction industry [10]. As additive particles in bitumen, these microcapsules must be triggered to break by microcracks and release a healing agent. The research content also includes the application proportion of microcapsules and a self-healing efficiency investigation [13].

The above research focused on the preparation and performance characterization of materials. This study mainly focuses on the preparation of microcapsules and the description of the self-repairing performance of bitumen. Due to the different preparation methods and the shell materials of microcapsules used by other healing agents, the additional number of microcapsules is also different, resulting in the performance of self-healing by microcapsules in bitumen. This leads to the above research contents being scattered and not systematic, and technical puzzles still need to be solved. So far, it can be summed up as two puzzles in the application of self-healing bitumen. One is that the microcapsule’s microstructure must meet the self-healing process requirements; while the mechanical properties of the shell can be improved, it is not easy to release the healing agent from the porous structure. On the contrary, microcapsule shells will not break with higher mechanical strength. The result is that a macrocrack may be triggered and lead to fracture without leaking the healing agent. Therefore, the shells are usually made of polymeric materials with appropriate strength and toughness. Another puzzle is how to maximize the self-healing efficiency of microencapsulated bitumen systems through a multiscale structural design and set up a mathematical model to predict the self-healing efficiency in bitumen after a long service time. Although the self-healing efficiency can be measured using mechanical tests, macroscopic prediction is still needed in which the factors of material structure parameters, temperature, and time are fully considered. It has been proved that the structure of self-healing bitumen is complex, and the molecular cross-linking degree determines the shell material’s strength. The tip stress of bitumen determines the number of microcracks penetrating the microcapsules [3]. The microcapsule rupture amount determines the amount of healing agent released [14]. The amount of repair agent determines the final self-healing capability. In self-healing, time and temperature determine the diffusion rate and diffusion range of healing agents from shells. However, a bridge has yet to be established between self-healing bitumen design, performance, and application effect. All the above research contents are fragmentary without any overall consideration. Establishing a genuinely reliable mathematical relationship between the material design and the actual application effect is impossible. This quantitative relationship should be fully understood at multiscale in order to design the material structure according to the requirements. Conversely, the mathematical relationship predicts a self-healing effect and feeds back the deficiencies of the material design from an actual real application.

At present, developing a simple, cheap, robust, and environmentally friendly method of designing self-healing microcapsules/bitumen is crucial for actual application in construction engineering. This work aimed to derive a mathematical model based on the parameters of the test system. This model considers the molecular size of the microcapsule wall material, the microsize of the wall material fracture, and the macrosize of the microcrack healing. The final self-healing effect estimation equation is built through the relationship between the various quantitative relationships in the self-healing details; this model will also consider the time and temperature factors in the self-healing process, which are significant for predicting the self-healing effect.

## 2. Self-Healing Mechanism Based on the Multiscale Structure

The self-healing process should be analyzed in detail based on the self-healing mechanism to establish the relationship between the molecular structure, microsize, and macrosize of the materials and understand the relationship between material structure and performance. In other words, the final quantitative relationship of the self-healing efficiency model can be set up only by understanding each step of self-healing and establishing the quantitative connection relationship between the self-healing steps. In pavement engineering, it has been found that liquid rejuvenator (healing agent) application is the only method that can restore the original properties of the pavements [2,6,11]. Usually, healing agents can restore and reconstitute the asphaltene/maltene ratio [10]. The healing agents have service restrictions because it is difficult to penetrate the pavement surface. In addition, the pavement must be closed for some time after their application because the liquid healing agents simultaneously cause a high reduction in pavement surface friction for vehicles. Interestingly, the usage of microencapsulation healing agents inside bitumen is a successful alternative approach. For example, Garcia et al. [15] fabricated capsules using sand as a skeleton and epoxy resin as a rejuvenator coating. Su et al. [4,5,10] reported a method to fabricate microcapsules with MMF resin shells containing rejuvenators for self-healing bitumen. These microcapsules met the requirements of self-healing bitumen, including reliable mechanical properties and satisfactory thermal stability [14]. Self-healing materials based on microcapsules which are structurally incorporated to repair damage caused by mechanical usage over time have been widely studied. Figure 1 illustrates the self-healing mechanism of bitumen using microcapsules containing a healing agent with multiscale steps of the molecular scale, microscale, and macroscale. In Figure 1a, a healing agent is microencapsulated by the shell (a_1_). The shell structure is formed with a certain thickness and strength (a_2_). The strength of the shell is usually determined by the cross-linking degree of the polymer (a_1_). These core-shell structural microcapsules are embedded within the bituminous materials, as shown in Figure 1b. When the microcrack extends to the outer shell of a microcapsule, the shell is punctured by the microcrack through the tip stress of the microcrack (b_1_). Several microcapsules may be broken simultaneously by microcrack propagation. Then the released healing agent seals the microcracks and permeates the surrounding bitumen, as shown in Figure 1c. The capillaries and penetration behavior will determine the self-healing efficiency of aged bitumen (c_1_, c_2_). With the help of capillarity, the healing agent flows into narrow microcracks without the assistance of, and in opposition to, external forces [10,16]. During the above self-healing process, several matters need attention and emphasis. The microcapsules’ core/shell structure design, such as size distribution, encapsulation rate, and non-biodegradability, needs special consideration because the structural design results will affect its performance. Generally, as the adhesive between aggregates, the thin layer thickness of bitumen is often less than 50 μm. Microcapsules exist in the above bitumen thin layer. To avoid being squeezed or crushed during asphalt molding, the size of the microcapsules containing the curing agent should be less than 50 μm. In addition, the shell thickness of the microcapsules must be controlled to ensure that microcapsules have excellent thermal stability and appropriate mechanical strength. It was found that a thicker shell could improve the mechanical strength of the microcapsules, but too high strength may result in the microcracks being unable to destroy these microcapsules [17].

## 3. Materials and Methods

### 3.1. Test Materials

Tiangong Tech Co., Ltd. (Tianjin, China) supplied the self-healing microcapsules as the commercial product in this work. The shell material was a commercial prepolymer of methanol-modified melamine formaldehyde. Three microcapsule types were named micro-A, micro-B, and micro-C, as listed in Table 1, which all had a core/shell weight ratio of 1/1. The average size of each microcapsule type had three size values of 10, 15, and 20 μm. The shell formation was carried out by cross-linking polymer molecules under pH values of 4–6. The shell thickness values of the three samples were 1.53, 1.54, and 1.54 μm. The healing agent was a commercial oil purchased from Aonisite Chemical Trade Co., Ltd. (Tianjin, China). The microcapsules’ core/shell weight ratio was 1/1, and the average sizes of the microcapsules were 10, 15, and 20 μm. The bitumen (80/100 penetration grade) was purchased from Qilu Petrochemical Industries Co. Ltd. (Linzi, China), and was manually processed through a thin film oven test into the aged bitumen (40/50 penetration grade).

### 3.2. Preparation of Self-Healing Bitumen Samples

Microcapsule particles were randomly mixed in the bitumen (80 °C, 300 rpm) with a weight ratio of 1, 2, 3, 4, and 5%. Then the mixture composite was placed in a rectangular silicone mold. The samples were put in a refrigerator at −10 °C for 24 h.

### 3.3. Observation of Self-Healing Microcapsules

An optical microscope was used to check the state of microcapsules in the self-healing process. The dried microcapsules were adhered to a double-sided adhesive tape without racking the shells. The surface morphologies were observed using a Philips L30 environmental scanning electron microscope (ESEM, Philips, Amsterdam, The Netherlands) at an accelerated voltage of 20 kV.

### 3.4. Testing Method of Self-Healing Efficiency (E_SH_)

The sensor recorded the initial tensile strength at break and registered it as (*S*_b0_). After the tensile fracture of the sample, it was placed in a 0 °C environment for five days according to the original sample morphology, and its tensile strength was tested again and recorded as (*S*_b1_). After the second self-healing of the sample in the same environment simultaneously, the tensile strength was tested again as (*S*_b2_). The E_SH_ values were calculated based on the rules shown in Equations (1) and (2),
(1)$$E_{SH\text{-}1}=\frac{S_{b1}}{S_{b0}}\times 100\%$$
(2)$$E_{SH\text{-}2}=\frac{S_{b2}}{S_{b0}}\times 100\%$$
where E_SH-1_ is the self-healing capability of the sample in the first self-healing cycle. E_SH-2_ is the self-healing capability of the sample in the second cycle.

## 4. Results and Discussion

### 4.1. Derivation of Model between Microscopic Scale and Macroscale

It is well known that bitumen is a polymeric-colloid mixture with an inherent self-healing capability. The decline of this capability is often caused by the aging caused by the loss of its small molecules. The molecule of the healing agent makes up for the loss of small bitumen molecules, and increases the softness of the asphalt. It is conducive to the movement, entanglement, and mutual fusion of bitumen molecules. This also increases the self-healing capability of aging bitumen. In detail, the diffusion rate of a small molecule of healing agent is affected by temperature, viscosity, and the aging degree of the bitumen. At the same time, when the healing agent molecules diffuse into the bitumen, the movement ability of the bitumen molecules is also greatly enhanced [10]. Through the interaction and entanglement of molecular chain segments, bituminous molecules can make cracks disappear quickly. At a specific temperature, a larger area of fracture interface needs more time to complete this process. This means that the self-healing ability cannot be rapidly improved. However, when more healing agents appear at the fracture interface, the movement ability of the bituminous molecules can be accelerated, which also virtually accelerates the self-healing efficiency of the material.

Based on the above analysis of the microcapsules’ bitumen self-healing mechanism, we know that three factors determine the self-healing efficiency (*E*_SH_) of the bitumen sample in the macroscale, that is, the *E*_SH_ value is a function of two variables, as shown in Equation (3),
(3)$$E_{SH}=f_{1}\left(V_{crack},\;\;m_{healing,\;\;\;}\varphi_{m}\right)$$
where *V*_crack_ is the volume of microcracks, *m_healing_* is the total amount of healing agent in microcracks and $\varphi_{m}$ is the mass percentage of microcapsules (wt.%). The shape and propagation state of the microcracks determines the volume of the microcracks. Meanwhile, it can be imaged based on the rheological principle of liquid, *m*_healing_ data are affected by the number of breaking microcapsules (*n*), temperature (*T*), and the zero-shear viscosity of the healing agent (*η_0_*_-healing_) and time (*t*). This *m*_healing_ value can be expressed as a functional relationship as Equation (4),
(4)$$m_{healing=f_{2\;\;}\left(m_{1}\cdot n,\;\;\;\eta_{0-healing},\;\;\;t,\;\;\;T\right)}$$
where *m*_1_ is the amount of healing agent contained in a single microcapsule. Here, it is assumed that the healing agent can be ultimately released from the damaged microcapsules. Generally, the healing agent is a small molecular organic compound, and the relationship between zero-shear viscosity and temperature can be expressed as Equation (5),
(5)$$\eta_{0-healing=𝜕\frac{\Delta E_{\eta }}{RT}}$$
where 𝜕 and *R* are constants, and Δ*E_η_* is the apparent activation energy of the healing agent. As *η*_0−healing_ is a function of temperature, Equation (4) can be simplified as Equation (6),
(6)$$m_{healing=f_{2\;\;}\left(m_{1}\cdot n,\;\;\;\;t,\;\;\;T\right)}$$

The diffusion behavior of healing agent molecules in bitumen is another factor to be considered in the self-healing process. Previous studies have shown that the self-healing process of bitumen is a periodic and continuous movement of a liquid restorative agent, including penetration, release, capillarity, and diffusion [10]. In general, Fick’s law can be used to describe diffusion behavior mathematically. The diffusion coefficient (*D*) is the molar flux and concentration gradient ratio. In this study, the aging degree, microstructure, and temperature of asphalt are the two main factors affecting the *D* value. After the healing agent flows out and penetrates, it diffuses into the aging bitumen material. The *D* value is determined by the penetration and diffusion process. In previous work [18], a preliminary mathematical model formula was given as shown in Equation (7),
(7)$$D=ae^{-e^{\left(b\left(T-T_{c}\right)\right)}}$$
where a, b, and c are constants. Although this rule may not be an accurate calculation, it still guides the design of the microstructure of shelled microcapsule agents. In addition, we can see from this relationship that temperature (*T*) is another crucial factor affecting *E*_SH_ value. In previous work [19], the movement of bituminous molecules depends on the time–temperature superposition principle. The *E*_SH_ model of bitumen can be described by the time–temperature superposition principle, as shown in Equations (8) and (9),
(8)$$E_{SH}={\left[1+{\left(\frac{\beta }{t\cdot \alpha_{T}}\right)}^{\frac{\log2}{n}}\right]}^{\frac{\delta }{\log2}}\cdot 100$$
(9)$$\log\alpha_{T}\left(T\right)=\frac{\Delta E_{a}}{2.303R}\left(\frac{1}{T}-\frac{1}{T_{0}}\right)$$
where α_T_ is the time–temperature superimposed displacement factor, *β* and *δ* are model parameters, Δ*E*_a_ is the apparent activation energy (unit: J/mol), and *R* is the general gas constant (8.314 J/(mol · K)). In this model, we have studied the time–temperature dependence of self-healing bituminous materials using microcapsules containing healing agents [20]. The *E*_SH_ value shows that increasing temperature within a specific range can enhance the self-healing effect of each healing cycle because higher temperature provides molecular motion energy, reduces molecular motion resistance, and accelerates the movement of the bituminous molecules and small molecules of the healing agent. According to the experience of the time–temperature equivalence principle, if the glass transition temperature (*T*_g_) of the polymer is taken as the reference temperature, the relationship between log(α_T_) and (*T* − *T*_g_) can be expressed by the WLF (Williams–Landel–Ferry) equation as shown in Equation (10). The temperature range of the sub-equation is (*T*_g_ to *T*_g_ + 100 °C) [20].
(10)$$lg\alpha_{T\;\;}=\frac{-17.44\left(T-T_{g}\right)}{51.6+\left(T-T_{g}\right)}$$

Based on the above analysis of the mechanism, the self-healing process is determined by two steps, namely, the penetration and diffusion of the healing agent. That is, the E_SH_ value is affected by the parameters of *V*_crack_, m_healing_, *D*, *T*, *η*, t and $\varphi_{m}$, which can be expressed as a function expression,
(11)$$E_{SH}=f_{3}\left(V_{Crack},m_{healing},D,T,\;\eta ,\;t,\;\varphi_{m}\right)$$

To sum up, we can give a functional relationship to describe the relationship between E_SH_ and the influencing factors, as shown in Equation (12). Equation (13) is the calculation formula of *m*_1_, where *r*_1_ is the average inner radius of spherical microcapsules and *ρ* is the density of liquid healing agent. In this study, microcrack propagation causes the fracture of the microcapsule’s shell material, and the tip-stress development presents a tearing extension mode. Therefore, a microcrack’s three-dimensional structure can be considered a conical shape with a lead. Its volume (*V*_crack_) formula can be approximately calculated as Equation (14), where *r*_2_ is the radius of the base area of the cone and *L* is the crack length.
(12)$$E_{SH}=f_{4}\left(V_{crack},m_{1}\cdot n,\;\;\;\;\;T,\;\;\;\;\;t,\;\varphi_{m}\right)$$
(13)$$m_{1}=\frac{4}{3}\pi r_{1}^{3}\cdot \rho$$
(14)$$V_{crack}=\frac{1}{3}\pi r_{2}\cdot L$$

To sum up, it was found that the main factor affecting E_SH_ value can be simplified as a function containing four variables: the size of microcracks with a macroscale (*V*_crack_), the number of broken microcapsules (*n*) with a microscale, time (*t*) and temperature (*T*). Because time and temperature are controllable factors, the conversion relationship between the two parameters can be established through the WLF equation. Combined with the above calculation Formulas (12)–(14), the final E_SH_ expression can be written as Equation (15),
(15)$$E_{SH}=f_{5}\left(r_{1},r_{2}\cdot L,\;\;\;\;\;T,\;\;\;\;\;\varphi_{m}\right)$$

The more complex and longer the crack grows, the more microcapsule breaks are needed to provide sufficient healing agent, as shown in Equation (16).
(16)$$\frac{4}{3}\pi r_{1}^{3}\cdot n\geq \frac{1}{3}\pi r_{2}^{2}\cdot L$$

In practical engineering applications, the size of microcapsules is generally the product size of fixed design. When the amount of healing agent released by the microcapsules just meets the needs of crack healing, Equation (16) can be simplified as Equation (17),
(17)$$r_{2}=\sqrt{\frac{4n}{L}\cdot r_{1}^{3}}$$
(18)$$E_{SH}=f_{6}\left(r_{1},\;L,\;n,\;T,\;\varphi_{m}\right)$$

After the above deduction, the conclusion was that the self-healing efficiency of the microcracks of bitumen is mainly determined by the length of microcracks (*L*), microcapsule radius (*r*_1_), and the number of broken microcapsules (*n*) as shown in Equation (18). At the same time, the ambient temperature (*T*) plays a role controlling the kinetic energy and vitality of molecular motion.

### 4.2. Derivation of Model between Microscopic-Scale and Molecular-Scale

In situ polymerization is usually used to prepare the self-healing microcapsules with a core-shell structure. Shell formation occurs in the continuous phase rather than on both sides of the interface between the continuous phase and the core material. Apart from the size, shell thickness, and shell macrostructure, the micromechanical character of the shells is another important property that needs to be considered [11,14]. The mechanical properties of the shells play an essential role in many processes, and an understanding is indispensable for their application [16]. Based on the material science principle, it can be imagined that the mechanical properties of a microcapsule are determined by shell structure, including molecular structure, molecular weight, shell size, and thickness. It has been reported that polymeric shell microcapsules are viscoelastic particles depending on size [14]. It is then clear that a thicker shell will increase the overall stiffness of a single microcapsule. However, producing a thicker shell may alter the bulk properties of the microcapsules in ways that are not easy to evaluate because, although essential, the membrane constitutive law is quite challenging to measure due to the smallness and fragility of artificial microcapsules. Hardness and Young’s modulus results indicated that the shell acted as a polymer with elastic–plastic deformation. The size and shell thickness are both main influencing factors of the micromechanical properties of the microcapsules. Because of the tiny structure, there is still little knowledge of the relationship between the micromechanical properties and the shells (size and thickness).

Apart from the size and shell thickness of the microcapsules, the cross-linkage chemical structure of the shell also influences its strength. The crosslinking density is the number of cross-linking bonds in the cross-linked polymer, which is generally expressed by the molecular weight of the network chain. The higher the crosslinking density, the more cross-linking bonds per unit volume, and the greater the degree of cross-linking. For crosslinked polymers used as plastics, such as epoxy resin, the higher the crosslinking density, the better the heat resistance and tensile strength. The higher the cross-linking degree, the lower the impact strength. For cross-linked polymers used as rubber, such as various rubbers, the cross-linking density is high, the mechanical strength is better, and the resilience is better. The polymer chains are connected into a three-dimensional space network macromolecule through branch chains, forming a cross-linked structure. The type of cross-linking bond and the cross-linking density are the critical parameters in the cross-linking structure, which indicate the structure of the cross-linking bond and the density of the cross-linking point between the molecular chains. The physical properties of the cross-linked polymer have significantly changed, and the properties most affected by the cross-linking density are modulus and hardness. Because the cross-linking point between the chains produced by cross-linking inhibits the sliding between the polymer chains, the modulus and hardness increase with the increase in the cross-linking density. The relationship between cross-linking density and tear strength is relatively complex in a range of cross-linking degrees; its performance has a peak value; after that, with the increase in the cross-linking density of the polymer compound, the tear strength gradually decreases.

Fracture mechanics believe that the stress intensity factor declines at the crack tip when it reaches the material’s fracture toughness, and the crack will expand; otherwise, the crack will not expand or stop growing. Therefore, accurately calculating the stress intensity factor is crucial when using fracture mechanics to study the crack growth problem. It has been reported that Griffith’s theory has been applied to analyze the condition of microcrack propagation from the energy perspective [19,20]. When the coarse elastic strain reduction in the object is more significant than or equal to the surface required for the crack to form two new apertures, the crack will expand. The conditions of microcrack growth can be analyzed from the energy point of view. However, it is complex and challenging to establish the mathematical model from the energy point of view for the self-healing bitumen composites. This problem can be solved from the point of view of the stress field at the tip of the microcrack. Fracture mechanics believe that when the stress intensity factor at the crack tip reaches the material’s fracture toughness, the crack will expand. Otherwise, the crack will not expand or stop growing. Therefore, when using fracture mechanics to study the crack growth problem, accurate assessment of the stress intensity of the material is essential. According to Griffith’s crack fracture mechanics theory [20], the critical stress of microcrack propagation (*σ*_c_) can be expressed as Equation (19),
(19) σc=2γsEπa12
where *r*_s_ is the surface energy per unit area of microcrack, *E* is the material’s elastic modulus, and *a* is half of the microcrack length (1/2 L).

It has been proved that when the shell material of microcapsules is subjected to external pressure, the elastic–plastic deformation process will occur [21]. When subjected to tip stress, it can be regarded as a tearing stress behavior. The critical break force of the microcapsule shell material (*σ*_b_) should be less than or equal to the acute propagation stress of the microcrack (*σ*_c_), the damage to the shell may occur as in Equation (20),
(20)$$\sigma_{b}\leq \sigma_{c}$$

Microcapsules are tiny particles with core-shell structures. The shell material can be regarded as a thin membrane structure. The stress of this membrane structure is a process moving from elastic deformation to plastic deformation. When the membrane material cannot bear the maximum deformation of an external force, the membrane will break immediately. To study the dynamic failure behavior of a spherical plastic shell under impact load, the deformation mode of the spherical shell is generally given by introducing isometric transformation. Then, the motion control equation of the large deformation dynamic failure of the shell under impact load is given using energy balance. The acting force, the relationship between the change in dent radius and center point displacement with time, and the maximum dent radius are given. This accurate micromechanical analysis method needs to be more convenient for estimating the damage to microcapsules and is not suitable for the micromechanical behavior of many microcapsules. According to the principles of fracture mechanics, the propagation modes of microcracks can be divided into three types: open mode, shear mode, and dead mode. From the stress fracture characteristics of asphalt, the fracture mode of cracks is mainly open-mode cracks caused by temperature change or aging. To simplify this research process, the breakage of microcapsules is regarded as puncture-break behavior under the open-mode force of bitumen. In this way, the shell material can be viewed as a layer of uniform membrane material, and the microcracks penetrate the microcapsule shell material through the strength of a puncture. Therefore, this complex problem can be simplified as a two-dimensional mechanical problem. The puncture-strength test is generally used to determine the material’s puncture or fracture characteristics. The puncture-strength test is generally a compression test. The test uses a probe or other device to compress the equipment until the material breaks or reaches its elongation limit. The true strength of the membrane increases with the increase in the cross-linking density. The puncture strength decreases when the cross-linking density increases again after reaching the maximum value. For cross-linked polymers, the average molecular weight (*M*_c_) between cross-linking points can be used to express the degree of cross-linking. In addition, the puncture strength often increases with the thickness of the membrane.

Figure 2 illustrates the microstructure scale of the shell and microcracks, the molecular scale, and the macroscale characteristics of bitumen. According to the material puncture rule, the damage energy of the microcapsule shell material (*G*_b_) can be expressed as the product of the puncture force (*σ*_b_) and the thickness of the shell material (*d*_s_) as shown in Equation (21). It is used to characterize the ability of the material to resist tearing, expressed by the tearing energy. The physical meaning of *G*_b_ is the energy consumed by the material puncturing per unit area. At the same time, for polymer membrane materials, when the degree of cross-linking is within a specific range, the puncture energy is linearly proportional to the degree of cross-linking. Therefore, for the small shell material of microscale in this experiment, we can approximately think that the *G*_b_ is proportional to its *M*c value as Equation (22) [20,21], ε and φ are constants., where r_2_/L is the sharpness of the microcrack. Sharp microcracks can easily puncture the shell material.
(21)$$G_{b}\propto \varepsilon \cdot \sigma_{b}\cdot d_{s}\cdot \frac{r_{2}}{L}$$
(22)$$G_{b}\propto \frac{\varphi }{\sqrt{M_{c}}}$$

Considering Equations (19)–(22) comprehensively, Equation (23) can be obtained through line-mathematical derivation containing the multiscales of the bitumen/microcapsule composite, where k is a constant.
(23)$$\left(\frac{1}{M_{c}}\right)\cdot \left(\frac{L^{3}}{d_{s}^{2}\cdot r_{2}^{2}}\right)=k\cdot r_{s}\cdot E$$

This equation includes the molecular scale parameter (*M*_c_) of microcapsule shell material, the microscale parameters of microcracks (*L* and *r*_2_) and the thickness of the microcapsule shell (*d*_s_), and the macroscale parameters of the bitumen materials (*r*_s_ and *E*). In this way, this equation builds a bridge between the different scale parameters. Establishing this mathematical model involves carefully considering complex problems such as polymer molecular structure, micromechanics, and composite material mechanics and establishing all parameter relationships, which would be a complex problem. However, there is an urgent need for a simple rough estimation in practical applications, which must be addressed. Therefore, considering this problem can appropriately simplify the influencing factors and build a simplified mathematical model. The relationship between the various parameters can be temporarily simplified as a linear relationship. Although this model differs from reality, it can be constantly improved, and optimization is the only way to solve the issue.

### 4.3. Simplified Mathematical Model Establishment of E_SH_

The ultimate purpose of self-healing microcapsule products is for their application to asphalt pavement to reduce microcracks caused by asphalt aging. Microcapsules generally exist in the bitumen matrix between aggregates, and the completion of self-healing is also realized in the bituminous material. Therefore, only the bitumen’s physical performance parameters are considered. When microcracks are triggered in bitumen, the final three parameters (*L*, *r*_2_, *n*) are random and uncontrollable. Variable values can be imagined such that each microcrack’s *L* and *r*_2_ values cannot be affected in the same way by the inhomogeneity of the material’s microstructure and the uncertainty of the microcrack direction. At the same time, the number of punctured microcapsules (*n*) is also inconsistent. The parameters in this paper can only be regarded as the average value approximately. It can be concluded from Equation (23) that only two of the influencing factors (*r_s_* and *E*) of the *E*_SH_ value are related to the physical parameters of bitumen. Where *r_s_* is the surface energy per unit area of microcrack, it is the non-volume work that must be carried out on the matter to reversibly increase the system’s surface area under constant temperature, pressure, and composition. Another definition of surface energy is the extra energy of the surface particles relative to the internal particles. This value relates to its softening degree for bitumen, determined by its aging degree att a certain temperature. When this value increases, the softening degree of the bitumen decreases, and its elastic modulus (*E*) increases. In general, the surface energy of solids is linearly related to their physical properties, such as elastic modulus and melting point, mainly because these physical quantities are closely related to the binding force between molecules in solids. Therefore, the two physical parameters of bitumen can have the following relationship as shown in Equation (24).
$$\left\{\begin{array}{cc} r_{s}=k_{2}\cdot E & \left(24\right)\\ r_{2}=\sqrt{\frac{4n}{L}\cdot r_{1}^{3}} & \\ \begin{array}{c} E_{SH}=f_{6}\left(r_{1},\;L,\;n,\;T,\;\varphi_{m}\right)\\ \left(\frac{1}{M_{c}}\right)\cdot \left(\frac{L^{3}}{d_{s}^{2}\cdot r_{2}^{2}}\right)=k\cdot r_{s}\cdot E \end{array} & \end{array}\right.$$

By combining Equations (17), (18) and (23) with Equation (24), we obtain Equation (25) through derivation and calculation. Parameters of *L*, *r*_2_, and n are uncontrollable, but *d*_s_ and *r*_1_ of the microcapsules related to these two variables are controllable.
(25)$$E_{SH}=f_{7}\left(M_{c},r_{1},d_{s,\;}E,\;T,\;\varphi_{m}\right)$$

By controlling the parameters of the microcapsules *(M*_c_, *d*_s_, *r*_1_ and $\varphi_{m}$), the E_SH_ value can be simulated by measuring the performance of bitumen (*E*) at a certain temperature. Through the above comprehensive analysis and mathematical deduction, the following two conclusions can be obtained considering the multiscale structure of self-healing bitumen:i.Under the conditions of constant temperature and bitumen material, the *E*_SH_ values can be determined by controlling the shell structure and the size of the microcapsule under a specific amount of addition.ii.The *E* value of bitumen can be obtained by measurement. When the microcapsules’ shell structure and size are fixed, *E*_SH_ values under different temperature conditions can be calculated according to the WLF equation.

### 4.4. Simplify the Influencing Factors through Experimental Data

In this experiment, we used three mature self-healing microcapsule products sold on the market (Tianjin Sinogo Tech. Co., Ltd., Tianjin, China). Figure 3a shows part of the microcapsule products (Micro-A_1_, Micro-A_2_, Micro-A_3_, and Micro-B_1_) with different sizes. At present, self-healing microcapsule materials can be produced on a large scale to meet the needs of practical applications. Microcapsules are a yellow powdery substance whose color is mainly determined by the color of the core material. Microcapsule powder (Micro-A_1_) was sealed in plastic packaging to prevent moisture absorption and contamination, as shown in Figure 3b. The pH values used in the shell-material synthesis of the three microcapsule products were 4, 5, and 6. Figure 3c–e show the optical microscope photographs of Micro-A_1_, Micro-B_1_, and Micro-C_1_ fabricated under polymerization pH values of 4, 5, and 6. Different pH values mean different M_c_ values for the shell materials. The core/shell weight ratios of the three microcapsule samples had consistent values of 1/1, and all shell thickness values were about 1.54 without any significant differences. It was reported [14] that the core/shell weight ratio mainly determines the shell thickness, and the microcapsule size does not affect the thickness value. The emulsification stirring speed controls the size of microcapsules. A faster stirring rate leads to a smaller size of microcapsule product [14]. In this work, each microcapsule type had three size values of 10, 15, and 20 μm. Figure 3f–h show the typical SEM morphology of Micro-A_1_, Micro-A_2_, and Micro-A_3_ with different diameter values. More details of the microcapsule surface can be observed on the enlarged SEM image (Figure 3i). Some microcapsule shells were deformed due to the elastic–plastic deformation of microcapsule shell materials (Figure 3j). The shell material of microcapsules can be regarded as a thin membrane material, as shown in Figure 3k,l, which can break or tear to a certain extent after being stressed. It is precisely because of this characteristic of the shell material that it is possible to realize the self-healing of bitumen.

According to Equation (25), the E_SH_ value can be simulated by measuring the performance of bitumen (*E*) at a specific temperature with various M_c_ values. In this work, all microcapsule products have nearly the same values; only *r*_1_ is a variable quantity. Bitumen is a typical viscoelastic material. The physical basis behind its viscoelastic behavior is that the molecular chain length of the hydrocarbons and their derivatives that make up bitumen is longer than that of conventional small molecules. The unique relaxation movement of the molecular chain makes asphalt temperature and time sensitive. According to the different modes of molecular motion, the behavior of bitumen can be divided into a viscous flow state, a viscoelastic state, and a glass state. The glass transition temperature (*T*_g_) is the characteristic temperature of the reversible transition of asphalt from a viscoelastic state to a complex and brittle glass state. A search of the literature found that the glass transition temperature of bitumen and its polymer-modified products is generally around −20 °C. Therefore, the temperature range investigated in this experiment is −20 °C to 30 °C. After the temperature exceeds 30 °C, the movement ability of bitumen molecules is strengthened, the material is softened, and its microcrack healing capability is improved.

Figure 4a–f illustrate the relationship between the various parameters of the self-healing process. In these figures, the ordinate is the self-healing efficiency (E_SH_) with data fluctuation during a simple situation considered as T = −20~30 °C, and the abscissa is the diameter of the added microcapsules. Microcapsule samples (Micro-A, Micro-B, Micro-C) with diameters of 10, 15, and 20 μm were separately mixed into an aged bitumen with a 2.0% additive amount forming three self-healing composite samples. These microcapsules were fabricated under pH values of 4, 5, and 6, and the final shell M_c_ was different. We can draw the following conclusions by observing these data and the data-change trend. Firstly, it was found that the temperature was the foremost important factor affecting the self-healing capability of bitumen. With the temperature increase, the E_SH_ values of the bitumen samples with the same amount and microcapsule type were significantly improved. For example, the bitumen samples with 2.0 wt.% microcapsules (10 μm, pH = 4) had E_SH_ values of 21.0% (−20 °C), 34.5% (−10 °C), 38.5% (0 °C), 42.1% (10 °C), 82.5% (20 °C), and 84.0% (30 °C), respectively. When at low temperatures, significantly below the glass transition temperature (*T*_g_), adding microcapsules could not effectively improve the self-healing capability of asphalt. The reason is that the healing agent released from the rupture of microcapsules could not rapidly penetrate the bituminous molecules. Bituminous molecules cannot move under low-temperature conditions to promote fusion between molecules. Secondly, when the added microcapsules had the same size at the same temperature, the cross-linking pH values of their shell materials were different, which could have had a better impact on the final *E*_SH_ values within the error range. The possible reason is that the shell material is a layer of thin membrane material with microsize. Compared with the macroscale and microcracks, the effect of the cross-linking of the shell material on the propagation stress of microcracks may be omitted in the practical evaluation process. When the microcrack propagates, the tip stress in the propagation is far greater than the puncture stress that the membrane shell material can bear. In addition, it was also found that the final E_SH_ values of the bitumen samples were unchanged under the same temperature conditions; even the additive microcapsules had the same pH polymerization value and different diameters. The curve connecting the data points presented the trend of a horizontal straight line.

### 4.5. Analysis of Addition of Microcapsules ($\varphi_{m}$) and the Modulus of Bitumen (E)

Through the above analysis, we found that the microstructure of the microcapsules does not play a decisive role in improving the final self-healing efficiency. On the contrary, the addition of microcapsules ($\varphi_{m}$), the modulus of bitumen (*E*), and the ambient temperature (*T*) are the decisive factors that ultimately affect the self-healing effect. According to a basic knowledge of polymer physics, the viscoelastic and modulus changes of amorphous polymers obtained at different temperatures conformed to the WLF equation (Equation (26)). *E*_0_ is the modulus of bitumen under a reference temperature of *T*_0_. Its modulus increases for specific bitumen material due to the loss of small molecules in the aging process. The loss of small molecules makes the bitumen harden, its deformability decreases, and its modulus increases accordingly. Because bitumen is a temperature-sensitive polymer, the modulus change caused by temperature change is much more significant than that caused by aging. In this study, it was simplified to consider only the influence of temperature on the modulus of asphalt materials. Therefore, the change in *E* values could be estimated by the WLF equation. In this way, only two factors need to be considered to study the E_SH_ values, namely, temperature (*T*) and additive mass percentage of the microcapsule ($\varphi_{m}$).
$$\left\{ \begin{array}{cc} \begin{array}{c} lg\frac{E_{0}}{E_{g}}=\frac{-17.44\left(T-T_{0}\right)}{51.6+\left(T-T_{0}\right)} \end{array} & \left(26\right)\\ E_{SH}=f_{8}\left(\varphi_{m},\;\;\;\;\;T\right) & \left(27\right) \end{array}\right.$$


Figure 5 shows the curves of the relationship between E_SH_ values and $\varphi_{m}$ under different temperature conditions with a self-healing time of 2 days. The structural parameters of the added microcapsules (Micro-A_1_) were r_1_ = 10 μm and d*s* = 1.53 ± 0.30 μm, and its shell polymerization condition was under pH = 4. According to the changing trend of these data, the following rules could be found: when the dosage of microcapsules is zero, the self-healing efficiency is very low at low temperatures (−10~−20 °C). But, at a relatively high temperature (30 °C), their self-healing efficiency is about 60%. This shows that the self-healing capability of bitumen is innate, and its self-healing behavior can occur when the molecules can move above the glass transition temperature. For bitumen samples with the same numbers of microcapsules, the self-healing efficiency increases rapidly with the temperature increase. This conclusion is consistent with previous research results. At the same temperature, the self-healing efficiency of bitumen samples increased with the microcapsule increase. From the slope of the curve, this increased rate was relatively low at low temperatures (−20~10 °C). However, when the temperature reached or approached the glass transition temperature of bitumen, the self-healing efficiency of the bitumen samples had a rapid growth trend with the increase in the microcapsules added at the same temperature. This can be seen from the slope of the data curve. The increase in numbers of microcapsules significantly increases the probability of microcracks penetrating the microcapsules during the growth process. The rise in numbers of damaged microcapsules also provides more healing agents, thus improving the self-healing efficiency. Another interesting phenomenon is that at the same temperature, when the content of microcapsules reaches about 2.2%, its growth trend slows down. As seen from the gentle curve, the self-healing efficiency was not be improved by increasing the addition amount. Our previous study calculated the addition amount according to the volume addition amount [10,11]. After the conversion to mass ratio, the conclusion was consistent with the previous research results. The results of other researchers may differ from the results of this experiment, which may be caused by the differences in the type of their microcapsules and the uniformity of their addition and mixing.

### 4.6. Influence of Time Factor

This study uses the WLF equation to simplify the influential parameters and is an essential theoretical guide for bitumen dynamic viscoelasticity. The above research shows that the additional numbers and temperature of the microcapsules are crucial factors affecting the self-healing efficiency of bitumen, and the time factor also needs to be considered. For the bituminous material applied to pavements, the external conditions it bears in the application process are very complex, and the service time of the material is very long. It is tough to study and predict the performance changes of materials over a large-scale time range. Therefore, the mutual conversion of time and temperature can be realized through the principle of time–temperature equivalence. The comparison of the actual test data with the derived data can verify the correctness of this theory and the rationality of simplifying the previous influencing factors.

Figure 6 shows the E_SH-1_ and E_SH-2_ obtained by the experimental tests over 20 days. To simplify the research, this work only studied the self-healing performance of bitumen samples with 1 and 2% microcapsules, based on the above research conclusion. The testing temperatures were 10, 20, and 30 °C near and above the glass transition temperature of bitumen. In addition, the primary data of the secondary-cycle self-healing efficiency of bitumen were also studied in this study; the above experimental data were calculated according to Equations (1) and (2). It was determined that the E_SH-1_ values were increasing under the same temperature over a prolonged time. Moreover, the bitumen had higher E_SH-1_ values with a relatively higher temperature during the same self-healing time. This result obeys the rule of the time–temperature superposition principle. Bituminous molecules have enough time to complete adequate movement to eliminate microcracks. In previous work, it was reported that self-healing bitumen using microcapsules had a multi-self-healing ability [16]. This work also investigated the time factor in a second cycle. The bitumen sample still had self-healing capability in the secondary self-healing cycle. However, the E_SH-2_ value was lower than the E_SH-2_ value under the same conditions. This conclusion consists of the reported results of multi-self-healing rules [11,16]. According to the self-healing mechanism, the self-healing process is based on the penetration of a healing agent. In the secondary cycle, the possibility of breaking microcapsules may be decreased. Under the same temperature, the E_SH-2_ value had nearly the same variation trend as E_SH-2_ values. Moreover, the curves of E_SH-1_ and E_SH-1_ were almost all straight-line change trends. The slope of the curve of E_SH-2_ values under the temperature of 10 °C was lower than that of the curve of E_SH-2_ values under the temperatures of 20 and 30 °C. This result also confirms that the E_SH_ values obey the principle of time–temperature equivalence.

### 4.7. Deviation Analysis and Future Work

In this work, several hypothesized methods were applied to simplify the mathematical analysis of the influencing factors of the self-healing efficiency of bitumen using microcapsules based on multiscale analysis. (1) The first one is that at the molecular scale, it was hypothesized that the damage energy of the microcapsule shell material (*G*_b_) value is approximately proportional to its *M*_c_ value. Moreover, the line-mathematical derivation containing the multiscale bitumen/microcapsule composite of Equation (24) may also ignore more details of molecules with a more complex relationship with the shell microstructure. (2) The second one was that on a microscale, the micromechanics of the broken shell were simplified. Its puncture stress was considered equal to the final tip stress of the microcrack in the extension process. This simplification is only partially reasonable. (3) The third one was that the WLF equation was applied at the macroscale to simplify bitumen samples’ E values. It could be imagined that *E* values may be significant with the WLF equation because of the mechanism characteristic of bitumen. Over service time, bitumen’s softening increases due to aging. In addition, various bitumen materials have different components leading to a significant deviation from the WLF equation. Although there may be many problems and errors, valuable results were obtained through the derivation based on the above-hypothesized methods. In future work, the micromechanical method should be applied to give more details of the broken shell. The cross-linking degree of the shell should be determined to understand the fracture mechanism of the shell. Moreover, a model should be developed to optimize the application of microcapsules in various bitumen materials. Finally, based on multiscale structures, a mathematical model will be supplied, including the influencing factors concerning the self-healing efficiency of bitumen using microcapsules.

## 5. Conclusions

In recent years, intelligent material applications in bitumen have been a hot topic in civil and material science and engineering. Self-healing bitumen, especially, using microcapsules has attracted more and more researchers to promote this technology’s progress. To date, no quantified mathematical derivation has helped in understanding the influencing factors of the self-healing efficiency of bitumen. Based on the above needs of basic research and engineering application, a mathematical analysis of the influencing factors of the self-healing efficiency of bitumen using microcapsules based on multiscale analysis was supplied in this paper. At the same time, experimental verification was also carried out to understand the main influencing factors. The following conclusions can be drawn:A mathematical analysis was carried out to understand the influencing factors of self-healing efficiency based on the self-healing mechanism of bitumen using microcapsules. From the viewpoint of multistructures, molecular size, microsize, and macrosize were fully considered, including structural dimensions, preparation conditions, and self-healing conditions.A mathematical equation was established as a bridge between the multiscale parameters, including the molecular scale parameter (*M*_c_) of the microcapsule shell material, the microscale parameters of microcracks (*L* and *r*_2_) and the thickness of the microcapsule shell (*d*_s_), and the macroscale parameters of bitumen materials (*r*_s_ and *E*).Based on the above analysis of the multiscale factor parameters, a more simplified mathematical study found that E_SH_ values could be determined by controlling the shell structure and size of the microcapsule under a specific amount of addition under a condition of constant temperature and bitumen material. Moreover, the E_SH_ values under different temperature conditions could be calculated according to the WLF equation. as the shell structure and size of the microcapsules were fixed.The experimental data test found that when the addition of microcapsules reached about 2.2 wt.%, the E_SH_ values of bitumen could achieve the maximum under the same conditions. Therefore, based on the actual experimental data and the results of the above mathematical analysis, it was finally found that the main factors affecting the E_SH_ values of bitumen are temperature and time.The experimental data confirmed that the bitumen’s self-healing process follows the polymer time–temperature equivalence principle. The self-healing effect may be improved by increasing the temperature and prolonging the time.

## Figures and Tables

**Figure 1 materials-16-05073-f001:**
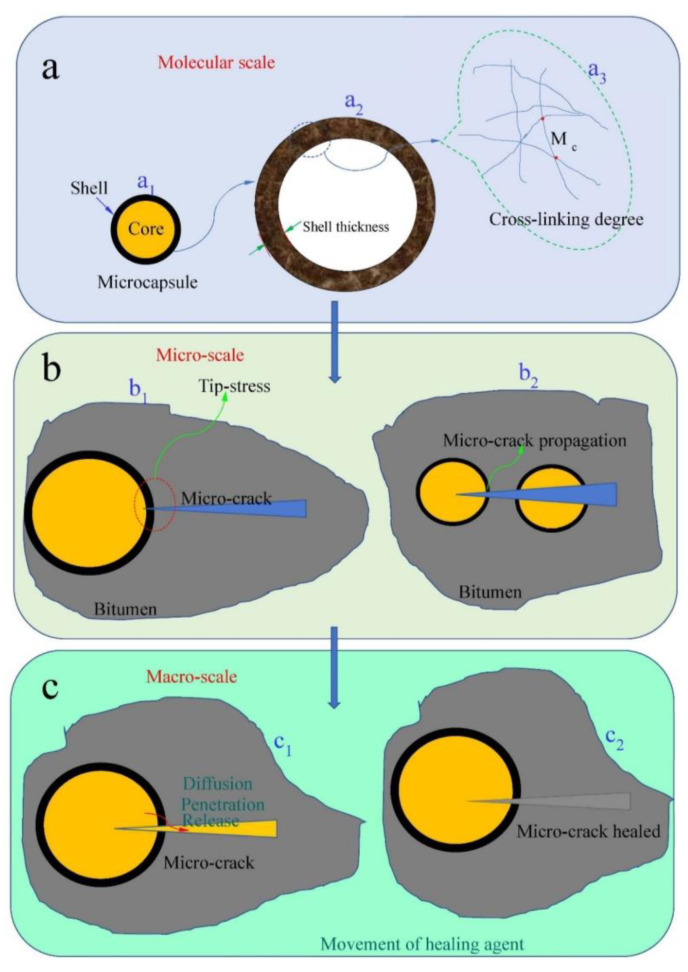
Illustration of the self-healing mechanism of bitumen using microcapsules containing healing agent with multiscale steps: from (**a**) molecular scale, (a_1_) microcapsule with core-shell structure, (a_2_) the polymer shell, (a_3_) the cross-linking degree of shell material; (**b**) microscale, (b_1_) the tip-stress of micro-crack, (b_2_) microcrack propagation, and (**c**) macroscale, (c_1_) diffusion, penetration and release of healing agent, (c_2_) micro-crack healed.

**Figure 2 materials-16-05073-f002:**
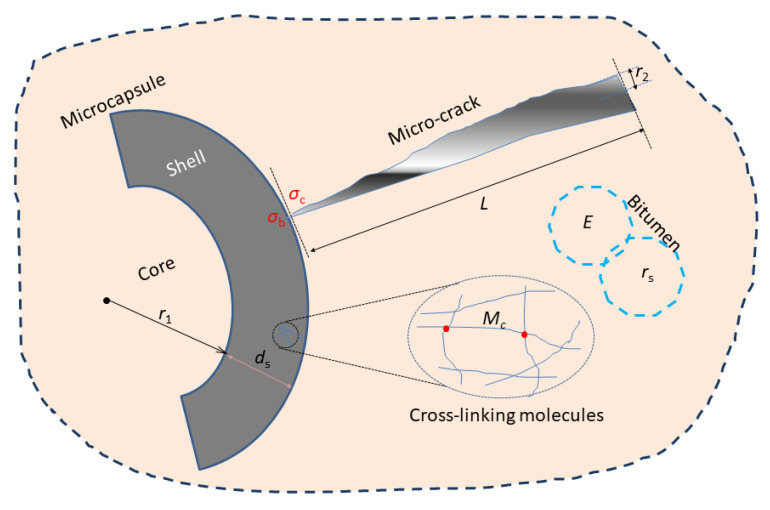
Illustration of the molecular scale of cross-linking shell, the microstructure scale of shell and microcrack, and macroscale characteristics of bitumen.

**Figure 3 materials-16-05073-f003:**
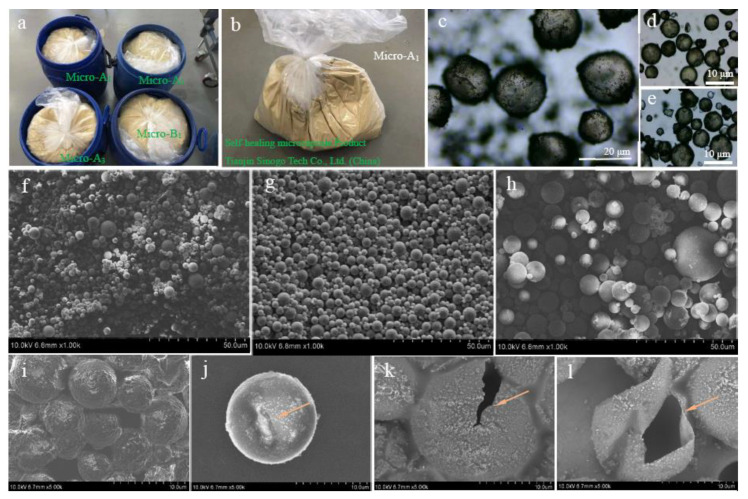
Self-healing microcapsule products. (**a**) Photograph of microcapsule products (Micro-A_1_, Micro-A_2_, Micro-A_3_, and Micro-B_1_) with different sizes; (**b**) microcapsule powder (Micro-A_1_) sealed in plastic packaging to prevent moisture absorption and contamination; (**c**–**e**) optical microscope photographs of Micro-A_1_, Micro-B_1,_ and Micro-C_1_ fabricated under polymerization pH values of 4, 5 and 6; (**f**–**h**) SEM morphology of Micro-A_1_, Micro-A_2_, Micro-A_3_ with different diameter values; (**i**) enlarged SEM morphology microcapsule surface (Micro-A_1_) with more detail; (**j**) single deformation of microcapsule shell; and (**k**,**l**) broken microcapsules.

**Figure 4 materials-16-05073-f004:**
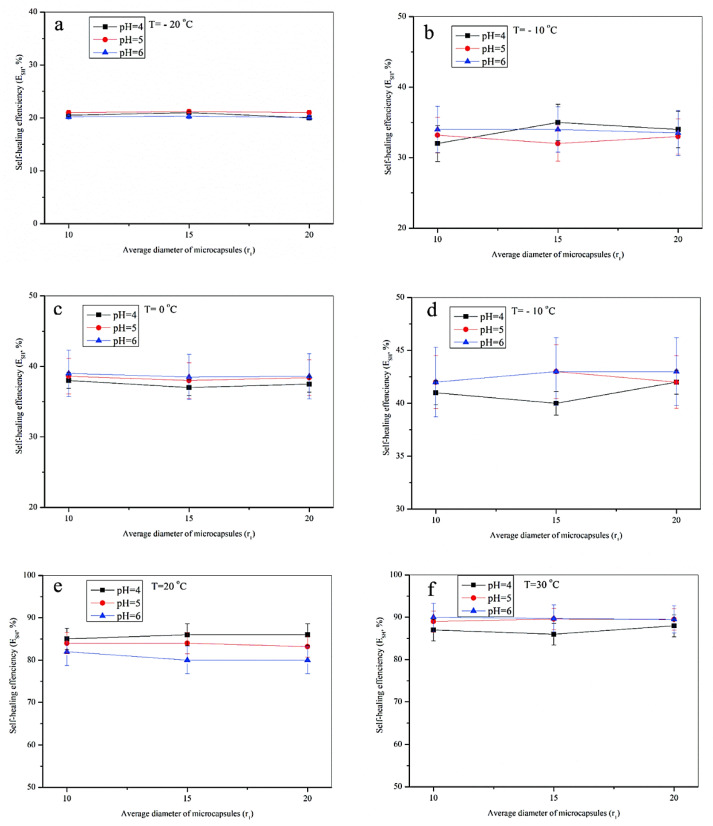
The relationship between E_SH_ values and various parameters of the self-healing process of bitumen samples, microcapsule samples (Micro-A_,_ Micro-B, Micro-C) with diameters of 10, 15, and 20 μm separately mixed into an aged bitumen with 2.0% additive amount, forming three self-healing composite samples, microcapsules fabricated under pH values of 4, 5 and 6, (**a**) −20 °C, (**b**) −10 °C, (**c**) 0 °C, (**d**) 10 °C, (**e**) 20 °C and (**f**) 30 °C.

**Figure 5 materials-16-05073-f005:**
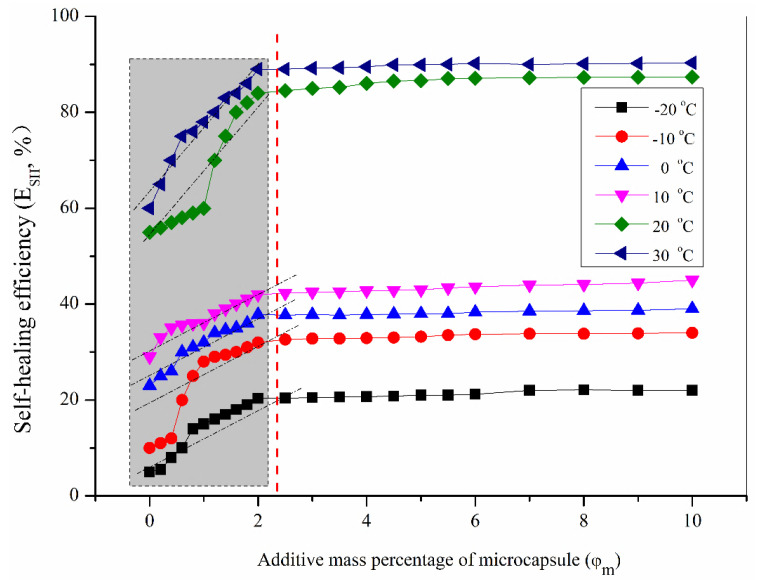
Curves of the relationship between E_SH_ values and $\varphi_{m}\;$ under different temperature conditions with a self-healing time of 2 days.

**Figure 6 materials-16-05073-f006:**
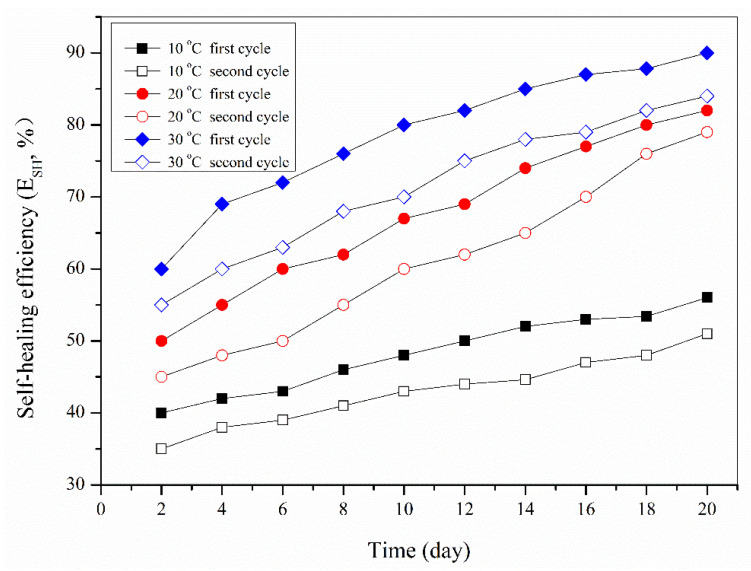
E_SH-1_ and E_SH-2_ values of bitumen samples with 2.0 wt% microcapsules at 10, 20, and 30 °C over 20 days.

**Table 1 materials-16-05073-t001:** Self-healing microcapsules with different structures.

Sample	Core/Shell Ratio	pH Value of Cross-Linking	Shell Thickness (μm)	Average Diameter (μm)
Micro-A	1/1	4	1.53 ± 0.30	A_1_ = 10, A_2_ = 15, A_3_ = 20
Micro-B	1/1	5	1.54 ± 0.26	B_1_ = 10, B_2_ = 15, B_3_ = 20
Micro-C	1/1	6	1.54 ± 0.27	C_1_ = 10, C_2_ = 15, C_3_ = 20

## Data Availability

Not applicable.

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
