# Peer review of "Multiscale Mathematical Analysis of Influencing Factors and Experimental Verification of Microcrack Self-Healing Efficiency of Bitumen Composites Using Microcapsules"

_materials, 2023, doi:10.3390/ma16145073_

Round 1

Reviewer 1 Report

Thanks to authors for considering Materials to submit their technical manuscript. Please, address the comments noted in the attached file.

Minor editing is required.

Reviewer 2 Report

Multi-scale mathematical analysis of influencing factors and experimental verification of micro-crack self-healing efficiency of bitumen composites using microcapsules

In the submission titled “Multi-scale mathematical analysis of influencing factors and experimental verification of micro-crack self-healing efficiency of bitumen composites using microcapsules” the authors have discussed above the self healing capacity of bitumen by using microcapsules in pavement engineering domain. The molecular size, micro-size, macro-size and the other factors that influence the self-healing capacity were considered. Mathematical analysis was done. The amount of healing agent released was considered to be most important than the volume of micro-cracks. The additive amount of microcapsule, temperature and time were the main factors considered. The study was concluded to have found a relationship between initial design and self healing.

The paper may be considered after revision and the authors shall address the following queries.

Major Points

1.      The authors have said in Page No. 3 that healing agents are service restrictions and re difficult to penetrate the surface and the pavement must be closed for a period of time since the liquid healing agents cause a high reduction in pavement surface friction for vehicles. Is there any such problem when using these micro-capsules?

2.      How does the healing agent flow into the micro-cracks (rheological behaviour) ?

3.      In case of too high strength microcapsules, how do micro-cracks appear and how they do not destroy the microcapsules?

4.      Why bitumen was pretreated using a thin film oven test?

5.      The authors shall include the number of times the tensile testing was done.

6.      How does pH value play a role in the behaviour of the micro-capsules?

7.      The paper is quite lengthy and if possible can be made shorter without affecting the essence of the study.

8.      How do higher temperatures influence the self-healing efficiency?

9.      At 20 °C , the self healing efficiencies converge than the others. How do the authors explain this point?

Minor Points

1.      In Page 2, Line No. 91, What does the “pours” mean? It may be “porous”. The authors shall clarify this.

2.      In Page No. 6, Line No. 127, it may be “acclerated voltage ‘of’ 20kV.

3.      In Page 6, Last line, what do the phrase “ and increases soften as asphalt “ refer to?

4.      The Line 338 – 346  and 351 – 358 in Page No. 10 are repeated. The repetition shall be removed.

5.      Similarly in the same Page No. 10, the lines 367 -  369  and 378 -380 are again the same content.

6.      In Page 13, Section 4.4 , Line No. 499 the first word “In” is missing.

7.      Expand  WLF.

English correction is necessary at some places and are pointed out. SOme repetitions are there.
